# Role of the TRP Channels in Pancreatic Ductal Adenocarcinoma Development and Progression

**DOI:** 10.3390/cells10051021

**Published:** 2021-04-26

**Authors:** Gonçalo Mesquita, Natalia Prevarskaya, Albrecht Schwab, V’yacheslav Lehen’kyi

**Affiliations:** 1Laboratory of Cell Physiology, INSERM U1003, Laboratory of Excellence Ion Channels Science and Therapeutics, Department of Biology, Faculty of Science and Technologies, University of Lille, 59650 Villeneuve d’Ascq, France; goncalo.mesquita@univ-lille.fr (G.M.); natacha.prevarskaya@univ-lille.fr (N.P.); 2PHYCELL—Laboratoire de Physiologie Cellulaire, INSERM U1003, University of Lille, 59655 Villeneuve d’Ascq, France; 3Institute of Physiology II, University Münster, 48149 Münster, Germany; aschwab@uni-muenster.de

**Keywords:** ion channels, TRP channels, pancreatic ductal adenocarcinoma

## Abstract

The transient receptor potential channels (TRPs) have been related to several different physiologies that range from a role in sensory physiology (including thermo- and osmosensation) to a role in some pathologies like cancer. The great diversity of functions performed by these channels is represented by nine sub-families that constitute the TRP channel superfamily. From the mid-2000s, several reports have shown the potential role of the TRP channels in cancers of multiple origin. The pancreatic cancer is one of the deadliest cancers worldwide. Its prevalence is predicted to rise further. Disappointingly, the treatments currently used are ineffective. There is an urgency to find new ways to counter this disease and one of the answers may lie in the ion channels belonging to the superfamily of TRP channels. In this review, we analyse the existing knowledge on the role of TRP channels in the development and progression of pancreatic ductal adenocarcinoma (PDAC). The functions of these channels in other cancers are also considered. This might be of interest for an extrapolation to the pancreatic cancer in an attempt to identify potential therapeutic interventions.

## 1. Introduction

Fifty years ago, the identification of a mutant fruit fly set the start to the discovery of a superfamily of channels [1]. The gene *trp* was only determined in 1989 and from then on, the TRP family started growing with several members being discovered in numerous independent studies [2]. In 2002 there was a standardization of the nomenclature that led to the formation of the superfamily TRP [3]. The TRP channel superfamily was divided into seven different groups based on sequence homology. For mammals, six subfamilies were reported: the TRPC or canonical family, the TRPV or vanilloid family, the TRPM or Melastatin family, the TRPP or polycystin family, the TRPML or mucolipin family, and the TRPA or ankyrin family (Figure 1). The channels belonging to these families have many diverse functions that include the sensation to pain, temperature, taste, pressure, and vision [4,5,6,7,8]. The role of these channels in diverse cancerous diseases has been established but there is a shortfall in the translation of this knowledge from bench-side to bed-side [9,10,11]. In pancreatic ductal adenocarcinoma (PDAC), there is a similar occurrence as there is still much work needed in order to better understand the possible roles of TRP channels and how can we take advantage of them clinically. The aim of this review is to summarise the data obtained so far on the expression and function of TRP channels in PDAC development and progression. It becomes evident that these channels have several functions in PDAC pathophysiology and could thereby be potential diagnostic and/or therapeutic targets for the disease.

## 2. Pancreatic Ductal Adenocarcinoma

PDAC is one of the deadliest cancers with the one of the worst outcomes, as the 5-year survival is at 9% [13]. Sadly, there has hardly been any improvement during the last five years. In 2015 the five-year survival was 7.5%. The patients which undergo resection or radical surgery have the best prognosis, but only less than 20% of the patients are diagnosed in a phase when the tumor is resectable [14,15]. The advances in the understanding of this disease have not yet led to an improvement of the patients’ outcome [16]. The PDAC microenvironment is very complex, consisting in an interrelationship between cancer cells, fibroblasts, epithelial, endothelial, and immune cells [17]. The formation of a dense stroma (that comprises nearly 90% of all tumor mass) is one of the hallmarks of this cancer [18]. This desmoplasia contributes to the development and progression of the cancer, either by secreting several factors that promote cell proliferation and matrix deposition [19], or by inducing immunosuppression in PDAC [20].

There is one aspect of PDAC pathophysiology that has not yet gained much attention. The development of this disease is also highly connected to Ca^2+^ signalling. Overall, Ca^2+^ has an important role in many cellular processes. The failure to regulate it can lead to known cancer hallmarks, like continued proliferation, tissue invasion and apoptosis resistance [21,22,23]. Ca^2+^ signalling can have a role in both cell proliferation and cell death. In proliferation, Ca^2+^ has a fundamental role for the initiation and progression of the cell cycle, as evidenced by the cell cycle arrest due to suppression of several Ca^2+^ channels [24].

The role of Ca^2+^ in apoptosis has also been assessed. Apoptosis can occur via the intrinsic or extrinsic pathway. Ca^2+^ plays an important role in the intrinsic pathway. In this pathway, there is an increase in the levels of Ca^2+^ inside the mitochondria which will lead to mitochondrial membrane permeabilization and the release of cytochrome c. This in turn will start a signalling cascade that culminates in the death of the cell [25]. Cancer cells can find ways to avoid the Ca^2+^-dependent apoptosis through modulation of Ca^2+^ levels inside the cell. For example, in breast cancer, overexpression of plasma membrane calcium-ATPase 2 helps controlling intracellular Ca^2+^ levels, conferring apoptosis resistance to the cells [26].

The migration processes are responsible for cancerous cell invasion into other tissues. Migrating cells exhibit Ca^2+^ gradients along their front–rear axis [27]. They are instrumental for directional migration. This Ca^2+^ gradient is either due to the activity of Ca^2+^ permeable channels in the plasma membrane or through the mobilization of Ca^2+^ from internal cell stores [28]. Furthermore, in order to migrate away from the primary tumor and invade surrounding tissues cells need to establish contacts with the extracellular-matrix (ECM). Ca^2+^ signalling leads to the disassembly of cell adhesions, with the cleavage of several focal adhesion proteins, through the Ca^2+^ sensitive protease calpain [29]. At the same time, Ca^2+^ signalling is also important for the formation of new adhesion points [30]. Ca^2+^ signalling can induce focal degradation of the ECM, through the upregulation of different matrix metalloproteinases [31,32]. The control and modulation of these Ca^2+^-dependent processes revolves around the TRP channels. They are both the signaller and the target of Ca^2+^-dependent processes and thus important players in Ca^2+^ modulated diseases, like cancer [33].

Furthermore, the specific PDAC tumor microenvironment might be impacted by TRP channels modulation. As previously stated, desmoplasia is a common feature in PDAC and consists of a dense layer of extracellular matrix (ECM) and fibroblasts, more known as cancer-associated fibroblasts (CAF) [34]. One of the cell types that can acquire this CAF phenotype are pancreatic stellate cells (PSC). These cells can contribute to the progression and metastasis of the tumor through secretion of cytokines and chemokines, raising of interstitial pressures and promotion of hypoxia [35,36,37]. TRP channels can be sensitive to all those stimuli or even participate in their signalling cascades (Figure 2).

## 3. TRP Channels

TRP channels are interesting due to the diversity of cation selectivity and specific activation mechanisms. All the channels possess a molecular structure similar to that of voltage-gated K^+^ channels, with six-transmembrane (S1 to S6) polypeptide subunits (with a pore-forming re-entrant loop between S5 and S6) (Figure 3A). They assemble as tetramers to form cation-permeable pores. Their functions vary strongly according to the intracellular domains (Figure 3B) [41,42].

## 4. TRPM

First described in 1998 by Duncan et al., the TRPM subfamily is the largest of the TRP superfamily [43]. The channels are quite diverse and can be grouped into four groups through phylogenetic analysis: TRPM1/3, TRPM2/8, TRPM4/5 and TRPM6/7 [44]. Several TRPM channels have roles in various cancers, but only TRPM2, 7 and 8 have been studied in PDAC. TRPM4 is connected to other cancers of epithelial origin, such as prostate cancer [45] and breast cancer [46]. Moreover, TRPM6 levels correlate positively with prolonged overall survival of colorectal cancer patients [47].

## 5. TRPM2

TRPM2 mutations have been associated with the survival time in patients with PDAC. Higher mRNA values of the channel correlate with lower survival times of the patients [48]. The in vitro results showed that TRPM2 is overexpressed in PANC-1 cells and that this overexpression results in enhanced cell proliferation and invasive ability. The authors also present a correlation between the expression levels of TRPM2 and the expression levels of ATP8B4, PARVG, TDRD9, TLR7, and SFMBT2. Of these, TLR7 seems to be the most important, as its expression is almost inexistent in healthy pancreas, but it is present in pancreatic tumor tissue and in pancreatic cancer cell lines [49,50]. TLR7 is implicated in the loss of expression of several players in cell cycle progression like PTEN, p21 or p27 [51]. Furthermore, SFMBT knockdown leads to higher levels of migration and invasion in prostate cancer cell lines [52]. This suggests that further research is required to investigate these associations. The mechanisms by which TRPM2 might impact PDAC progression is far from being understood. It was shown to have a role in promoting the expression of inflammatory cytokines by PDAC cells and enhancing their migratory capacity by importing Ca^2+^ in a pathway which sees SIRT6 modulating intracellular ADPr levels [53].

## 6. TRPM7

The study of this channel in pancreatic cancer started in zebrafish with a report stating that TRPM7 is functionally required for pancreatic epithelial proliferation [54]. In 2012, Rybarczyk and his colleagues showed that TRPM7 is present in the human pancreatic tissue and involved in magnesium-dependent processes of cell migration. They also showed that this channel is overexpressed in PDAC and inversely correlates with patient survival, which was further confirmed by Yee et al. [55,56]. Additional studies suggest a regulation of MMP expression by TRPM7. Indeed, a correlation between TRPM7 and Hsp90/uPA/MMP-2 pathway was demonstrated, as the silencing of the cation channel in different pancreatic cell lines leads to significantly decreased levels of the proteins of the pathway [57]. More recently, a study demonstrated that TRPM7 interacts with the ribosomal protein SA to promote MIA PaCa-2 cell migration stimulated by elastin-derived peptides [58].

## 7. TRPM8

The first TRPM channel being reported in pancreatic cancer was TRPM8. In 2010, much like TRPM7 and TRPM2, TRPM8 was reported to be overexpressed in pancreatic cancer cell lines (PANC-1 and BxPC-3) when compared to pancreatic ductal epithelia [59]. Furthermore, TRPM8 was required for maintaining cell proliferation. The silencing of this channel leads to morphological changes at cellular and nuclear levels, suggestive of replicative senescence. The silencing of TRPM8 also leads to an increased proportion of cells in G0/G1 phases and a decrease in S and G2/M phases. The aberrant expression of TRPM8 in humans was later confirmed [60]. The channel is required to prevent non-apoptotic cell death that involves replicative senescence of cancer cells. More studies have arisen to understand the possible role of TRPM8 in this type of cell death. For example, in breast cancer cells, TRPM8 can enhance basal autophagy by positively regulating AMPK [61].

In addition, TRPM8 expression could have a positive correlation with cancer development and metastasis formation. In BxPC-3 and MIAPaCa-2 cell lines, TRPM8 is crucial for the invasive ability of the cancer cells [62]. Furthermore, a comparison of 110 pairs of pancreatic cancer tissues and adjacent non-cancerous tissues revealed a correlation of higher expression levels of TRPM8 with worse overall survival and disease-free survival [63]. On the other hand, studies in PANC-1 cells showed that the silencing of TRPM8 stimulates cell migration and proliferation levels [64,65]. Through comparison between glycosylated and un-glycosylated TRPM8, using tunicamycin (an N-glycosylation inhibitor), Ulareanu et al. demonstrated that the un-glycosylated form may have a protective role in PDAC and that this is the form of the channel found in PANC-1 cells [65].

TRPM8 targeting can also be used to enhance the effects of gemcitabine, a common drug used in chemotherapy, in vitro [66]. Combining the silencing of TRPM8 and gemcitabine treatment elicits a more pronounced inhibition of cell proliferation and migration in BXPC-3 and PANC-1 cell lines. This might be explained by the down-regulation of multidrug resistance-related factors, P-gp, MRP-2, and LRP, that occur in response to TRPM8 silencing [66]. These proteins are responsible for cellular efflux of xenobiotics including gemcitabine. New cancer therapies seek the modulation of these proteins in order to achieve higher treatment efficacy [67].

## 8. TRPV

The Vanilloid subfamily of TRP got its name in 1997 because one of its members, TRPV1, was shown to be activated by the vanilloid compound capsaicin [68]. This subfamily is divided in two groups based on sequence homology; TRPV1 to 4 and TRPV5 and 6. The first four channels are all responsive to high temperatures (from 25 °C, in TRPV4, to 52 °C, in TRPV2) and have nonselective cation conducting pores. TRPV5 and 6 are the most Ca^2+^-selective TRP channels. TRPV2 has an important role in the pancreatic endocrine system, more specifically in the process of insulin secretion [69]. Although there is no information on this channel regarding PDAC, TRPV2 has been connected to various types of carcinomas. TRPV2 levels are higher in prostate cancer and in human hepatoblastoma, while being lower in urothelial carcinoma [70,71,72]. Furthermore, it promotes the progression of prostate cancer to more aggressive stages [70]. The presence of TRPV2 in both ductal and acinar cells might be of interest in the study of PDAC [69,73].

## 9. TRPV1

In 2006 studies connected the role of TRPV1 channels to pancreatic cancer. Hartel et al. demonstrated the overexpression of the TRPV1 in patients with both PDAC and chronic pancreatitis. This overexpression was higher in PDAC than in the chronic pancreatitis. Moreover, there was a positive correlation between pain levels and expression of TRPV1 in PDAC patients [74]. More recently, TRPV1 was found to be an important regulator of EGFR expression [75]. EGFR has been known for a long time as an interesting target in the pathology of cancer in general and PDAC, in particular [76]. It has a relevant role in the development and progression of the disease, like its interaction with the KRAS oncogene or the signalling induction for pancreatic cancer cell migration [77]. Overexpression of TRPV1 (or a TRPV1 agonist) downregulates EGFR expression and its signalling. The opposite occurs upon silencing of TRPV1 [75].

Another important program in PDAC progression is epithelial-to-mesenchymal transition (EMT). One of the EMT activators is Zeb1. Krebs et al. demonstrated that the depletion of Zeb1 leads to reduced invasion, distant metastasis and colonization capabilities in KPC mice [78]. These mice possess a mutation in the *Kras* and *Trp53* genes, which allows the study of PDAC in immunocompetent animals, while experiencing several features observed in human PDAC. TRPV1 might be controlling Zeb1 in pancreatic cancer as it has been shown in hepatocellular carcinoma. The knockout of TRPV1 channels in mice leads to an upregulation of Zeb1 and subsequent promotion of hepatocarcinogenesis [79]. So far, it is not known whether such a link between TRPV1 and Zeb1 also exists in PDAC.

## 10. TRPV4

Chronic pancreatitis is a major risk factor for the development of PDAC and other pancreatic cancers. The risk of PDAC in a chronic pancreatitis patient is elevated 16-fold [80]. Chronic pancreatitis can develop among others from alcohol, smoking, genetic predisposition, and autoimmunity. Some lesions that occur in pancreatitis (like acinar-ductal metaplasia, activation of PSCs leading to fibrosis) are present in Pancreatic Intraepithelial Neoplasia, which can further evolve into PDAC [81,82].

A correlation between high fat/alcohol and TRPV4 was found in pancreatic stellate cells from Lewis rats [38]. The high fat/alcohol diet leads to increased cytosolic Ca^2+^ mobilization in PSCs which is TRPV4-dependent. Furthermore, this process is enhanced by TNF-α. TRPV4 might be an important player in the development of PDAC from pancreatitis as it modulates Ca^2+^ mobilization in stellate cells and is up-regulated in pancreatic cancer tissues [83]. More recently, a report demonstrated that pressure-induced pancreatitis is promoted by Piezo-1 [84]. The continuous activation of this channel induces Ca^2+^ overload which leads to mitochondrial depolarization and trypsinogen activation in pancreatic acinar cells [85]. TRPV4 is activated by Piezo-1 so that the use of TRPV4 receptor antagonists inhibits the sustained Ca^2+^ elevation. In a mouse model of pancreatitis due to pressure (by pancreatic duct obstruction), TRPV4-KO mice are significantly more protected [85]. This relationship between TRPV4 and Piezo-1 might not be strictly linked to stellate cells. Stress-induced deformation of immune cells occurs in both pancreatitis and PDAC. This occurs as a result of the high interstitial pressure found on these affected tissues and might be activated by cell responses Piezo-1 dependently [36,86].

## 11. TRPV6

TRPV6 is one of the newest TRP channels studied in the context of PDAC. However, it has several connections to the pancreas. This channel has been described to regulate proliferation of INS-1E cells, which are a model of pancreatic β-cells, and BON-1 cells, which are a model of endocrine tumor cells [87,88]. Furthermore, increasing evidence suggests that the overexpression of TRPV6 is a common event in cancers of epithelial origin, such as ovary, prostate, thyroid and colon [89,90,91,92]. This subject can be controversial in PDAC as reports have presented contradictory information. While Zaccagnino et al. found a reduced expression of TRPV6 channels in micro-dissected PDAC samples [93], Song et al. reported an overexpression in PDAC tissues [94]. However, both of these studies did not take into account whether the tissue samples were from invasive or non-invasive parts of the tumor. This is a relevant distinction, as a preponderance of TRPV6 expression was shown for the invasive parts of breast cancer [89]. The silencing of TRPV6 in Capan-2 and SW1900 cells leads to decreased proliferation, increased apoptosis, lower levels of migration and invasion and cell cycle arrest. A phase I clinical trial for SOR-C13, a TRPV6 inhibitor, has been made with some positive results [61]. This trial had included 23 patients with different types of advanced solid tumors, two of whom had PDAC. Stable disease was observed in both patients. One of the patients had a 27% reduction from baseline in the sum of tumor diameters (by RECIST criteria), with a decrease of 55% in tumor marker CA19-9 [95]. TRPV6 levels also correlate with lower survival rates on patients with PDAC [94]. Adding to the fact that TRPV6 is also linked to early onset chronic pancreatitis, a risk factor for the development of PDAC, makes it indispensable to undertake a more detailed analysis of TRPV6 channels in pancreatic cancer [96]. 

## 12. TRPML

The TRPML subfamily is represented in mammals by TRPML1, 2 and 3 (also known as MCOLN-1 to 3). Mutations on TRPML1 or TRPML3 can lead to several disorders such as Mucolipidosis type IV (TPLM1) or deafness and pigmentation defects (TPLM3). On the other hand, no disease phenotype has been associated with TRPML2 [97]. Furthermore, this channel has a reduced presence in the pancreas [98].

## 13. TRPML1

TRPML1 is the founding member of the mucolipins subfamily of TRP. In pancreatic cancer this channel might predict poor survival of the patients with PDAC. In a study with 80 PDAC patients, high levels of TRPML1 are associated with a short overall survival and relapse-free survival compared to the patients with lower levels of the channel. Furthermore, downregulating the expression of TRPLM1 in BxPC-3 and PANC-1 pancreatic cancer cell lines leads to lower proliferation levels. When transfecting PANC-1 cells into the mice, the tumors have significantly less volume when the TRPML1 channel was silenced [99].

## 14. TRPML3

TRPML3 belongs to a set of nine genes whose signature was identified in an in-silico analysis to predict overall survival of PDAC patients [100]. Based on the role of TRPML3 in autophagy in HEK293T cells [101], the authors proposed that TRPML3 might exert its tumor suppressive effects in PDAC by this mechanism, too. So far, functional data are missing.

## 15. TRPC

This subfamily consists in seven members (TRPC1-7), all expressed in humans, except for TRPC2, which is a pseudogene. Although these channels contribute to the progression of several different cancers, their function in pancreatic cancer cells has not yet been studied in greater detail. However, a role of this subfamily has been described for pancreatic stellate cells.

## 16. TRPC1

The role of TRPC1 in migration of PDAC cell lines (BxPC-3) was investigated [102,103]. Firstly, in BxPC-3 cells, TGF-β could induce PKC-α through its translocation to the plasma membrane. This activation could be the result of the mobilization of intracellular Ca^2+^ by TGF-β. Furthermore, they confirmed that TGF-β can induce PTEN suppression and increase cell motility PKC-α dependently [102]. The only TRPC channels present in BxPC-3 are TRPC1, 4, and 6. Of these three, co-silencing of TRPC1 and NCX1 was able to inhibit TGF-β induced cell motility [103]. On another note, Fels et al. suggested that TRPC1 channels contribute to the pressure-induced activation of pancreatic stellate cells [39,104]. Using primary stellate cells from mice, it was observed that the knock-out of TRPC1 leads to significantly lower levels of cell migration after pressure incubation. Cell migration was used as a functional read-out of pancreatic stellate cell activation. Furthermore, the loss of TRPC1 channels causes a reduced Ca^2+^ influx after pressure incubation. These differences were not noticed under control conditions [39].

## 17. TRPC3

Not much is known about the role of TRPC3 in PDAC, but a recent study in RLT-PSC (pancreatic stellate cell line) opened the discussion on this channel. TRPC3 plays a role in the migration of stellate cells. It does so through an interplay with the Ca^2+^ sensitive K^+^ channel of intermediate conductance, K_Ca_3.1. Moreover, the suggested hypothesis is that the K_Ca_3.1 takes advantage of Ca^2+^ entering the cells through TRPC3 channels to activate calpain and promote deadhesion of the stellate cells [29].

## 18. TRPC6

As observed with TRPC1 and TRPC3, TRPC6 also modulates pancreatic stellate cell behaviour [40]. In primary murine pancreatic stellate cells, the knock-out of TRPC6 leads to impaired cell migration when the cells were exposed to a hypoxic environment. Pre-treating the cells in a hypoxic environment or inducing chemical hypoxia with dimethyloxalylglycine (DMOG), leads to increased migratory activity of wild-type cells but not of TRPC6 knock-out cells. Similarly, hypoxia increases Ca^2+^ influx and cytokine secretion in a TRPC6-dependent manner in pancreatic stellate cells. These TRPC6-dependent processes can lead to a continuous autocrine activation of pancreatic stellate cells and PDAC progression [40].

TRPC6 channels are also crucial for CXCR2-mediated responses of neutrophils. This has a great importance due to the role of neutrophils in the development of PDAC [105]. Neutrophils are part of the desmoplastic tumor microenvironment and a high density of intratumoral neutrophil granulocytes is associated with poor patient survival. Their recruitment needs CXCR2 signaling, which is TRPC6-dependent [106].

## 19. Targeting TRP Channels in PDAC

The current treatment of PDAC varies depending on the phase the cancer is diagnosed. Surgery is the still the only way to cure PDAC, but that it is a choice for patients diagnosed with resectable tumors only. Such an early diagnosis only occurs in ~20% of the cases [107,108]. This calls for better diagnostic tools and for a bigger array of biomarkers for PDAC. The TRPM channels appear to be overexpressed in PDAC. TRPM2 and 7 are overexpressed in humans while TRPM8 is reported overexpressed in pancreatic cancer cell lines. Further studies should focus on understanding on which PDAC stage these channels start being overexpressed, in order to understand the potential for these channels as diagnostic markers. For later stages of PDAC, the commonly used treatments are FOLFIRINOX and nab-paclitaxel–gemcitabine [107]. One example of a possible targeted therapy in the silencing of TRPM8, which enhances the effects of gemcitabine in vitro, in pancreatic cancer cell lines [66]. Nonetheless, the patients’ average time of survival in these later stages is below 12 months. The future strategies might involve more co-adjuvants that lead to better efficacy of the current drugs. The desmoplastic stroma is a hindrance to the development of more effective treatments for PDAC. Much of the chemo-resistance and high interstitial pressure derives from the desmoplasia processes. Furthermore, stomal cells like PSCs are crucial in the development of PDAC. Pothula et al. presented a novel therapeutic approach to PDAC by targeting a factor secreted by PSCs [109]. This treatment, in mice, managed to surpass the efficacy of gemcitabine in reducing tumor angiogenesis and metastasis. TRP channels could be a target to control the activation of PSCs and prevent the continuous stroma-tumor interactions. Both TRPV4 and TRPC1 are possible targets due to their role in PSCs pressure-induced activation [39,85]. A therapy with focus on TRP channels could be the key to augment the efficacy of current treatments by focusing the specific tumor microenvironment present in PDAC.

Nonetheless, to this day, no pharmacological agents targeting TRP channels have succeed in becoming a normalized treatment. A phase I clinical trial has been undertaken for an antagonist for TRPV6 in solid tumors. As mentioned above, this trial demonstrated that the use of SOR-C13 leads to reduced tumor size and CA19-9 marking. This treatment was made on patients with advanced tumors. One of the biggest obstacles to the development of TRP-related therapies might be to find specific inhibitors for each TRP channel. Not all TRP channels possess a specific antagonist like TRPM3 [110] or TRPV4 [111]. Some antagonists have achieved phase I clinical trials, like AMG 517 (TRPV1 antagonist), but ended up failing, as the treatment in humans elicited long-lasting hyperthermia [112].

For palliative care, a targeted therapy against TRPV1 might be of interest. As stated before, the intensity of pain reported by PDAC patients is correlated with the expression of TRPV1 [74]. Using an in vivo bone cancer model, Ghilardi et al. demonstrated the potential in targeting TRPV1 with an antagonist (JNJ-17203212) [113]. Mice with TRPV1 knockout and TRPV1 antagonist experienced a significant attenuation in pain-related behaviors in comparison to wild-type animals [113].

## 20. Conclusions

Despite of multiple efforts to understand and treat PDAC, the overall survival of the patients has not been vastly improved. Nonetheless, a lot of data has been surging in the past years. Taking advantage of that, in the last decade, the TRP channels have been appearing in a diversity of scientific works concerning their expression in the tumor tissue and the possible effects that they exert on the development and progression of PDAC. Almost every TRP subfamily present in humans seems to play a role in this disease. These ion channels might be a crucial tool, not only in the diagnosis of PDAC, but also as a therapeutic target that can be combined with traditional treatments. Numerous possibilities might arise from further studies on the TRP family, as there are still many unanswered questions. How can we translate this knowledge from bench to bed-side? There is a gap in our understanding of redundancy among these channels. Several studies demonstrated that the knockdown of a TRP channel leads to the upregulation of other TRP channels in order to balance it [39,114,115]. How can we circumvent these mechanisms to achieve better efficacy in treatment? Furthermore, a better understanding on the relationship between these ion channels and the immune system is pivotal as demonstrated by the role of TRPC6 in neutrophil chemotaxis [106]. Lastly, the increasing attention given to the stroma cells should be correlated with the TRP superfamily. Several of these channels have an impact on stellate cell migration and activation. Further study is needed to better understand the possible roles of TRP channels in the development of the specific tumor microenvironment found in PDAC.

The critical need for better diagnosis tools and new therapeutics might have a solution in this big family of ion channels that can affect both pancreatic cancer cells and pancreatic stellate cells and have roles in cell proliferation, migration, invasion, and death, as summarized in Figure 4.

## Figures and Tables

**Figure 1 cells-10-01021-f001:**
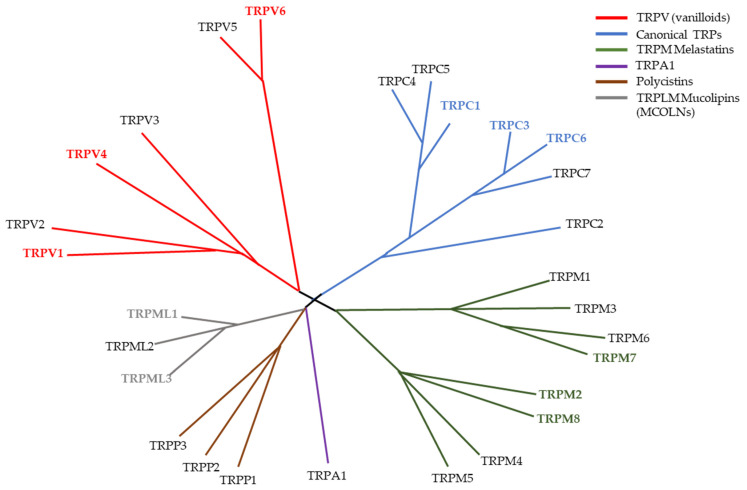
Phylogenetic tree of the TRP superfamily in vertebrates. Highlighted TRP channels (TRPV1, TRPV4, TRPV6, TRPC1, TRPC3. TRPC6, TRPM2, TRPM7, TRPM8, TRPML1 and TRPML3) identify channels that already been study in the context of PDAC and/or pancreatitis. Figure inspired by Clapham D. (2003) [12].

**Figure 2 cells-10-01021-f002:**
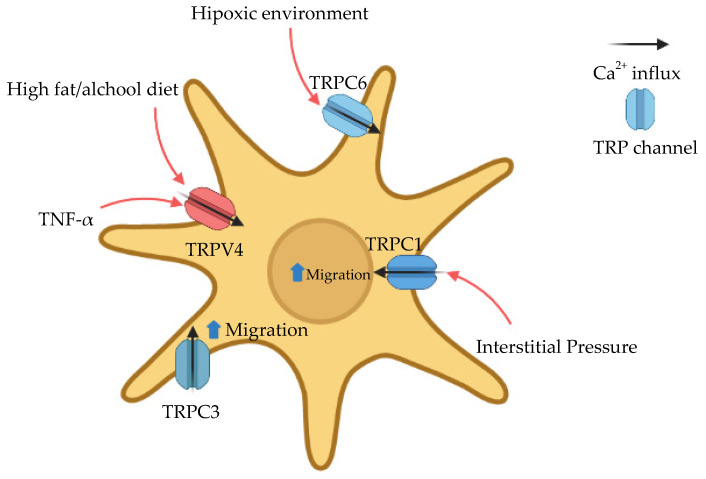
Ca^2+^ modulation by TRP channels in pancreatic stellate cells. These cells are important for the formation of the desmoplasia commonly found in PDAC [35,36,37]. Several reports have shown the impact that TRP channels might have on the activation and migration of stellate cells [29,38,39,40] This could bring to light potential therapies involving TRP channels and pancreatic stellate cells, to suppress the desmoplastic reaction.

**Figure 3 cells-10-01021-f003:**
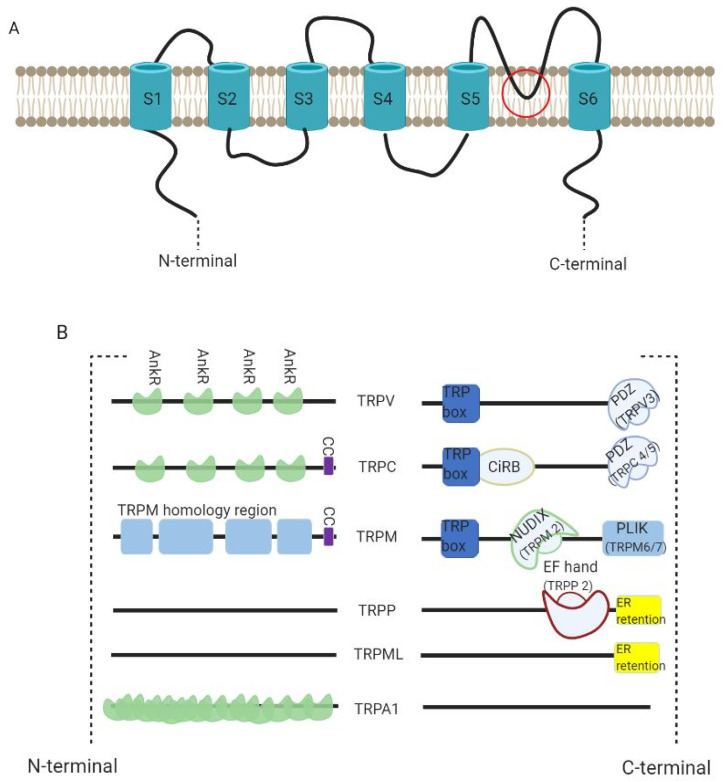
TRP channel structure. (**A**) All the channels possess a six-transmembrane (S1 to S6) polypeptide subunits (with a pore-forming re-entrant loop between S5 and S6 as marked in the figure) that assemble as tetramers to form cation-permeable pores. (**B**) The diversity of the channels is dependent on their C- and N- termini. All TRPV channels have a TRP box at their C-termini and TRPV3 has an additional PDZ binding motif. They have 3–4 Ankyrin repeats (AnkR) in their N-termini. TRPC channels have a TRP box containing the motif EWKFAR plus CIRB (a calmodulin- and inositol triphosphate receptor-binding site). Much like TRPV3, TRPC4/5 have a PDZ binding motif. This subfamily has 3–4 AnkR with an additional coiled-coil domain (CC) in the N-terminus. TRPM channels also have a TRP box in their C-termini. While TRPM2 has NUDIX (a NUDT9 hydrolase protein homologue binding ADP ribose), TRPLM6/7 have PLIK (a phospholipase-C-interacting kinase). In their N-termini, TRPM channels have a CC and a homology region whose functions are unknown. Both TRPP and TRPML have an endoplasmic reticulum retention signal (ER retention) in the C-terminal. TRPP2 also possess a EF-hand (a canonical Ca^2+^-binding domain) in the N-terminus. TRPA1 channels have a much bigger number of AnkR than TRPV or TRPC. Figure inspired by Clapham D. (2003) [12].

**Figure 4 cells-10-01021-f004:**
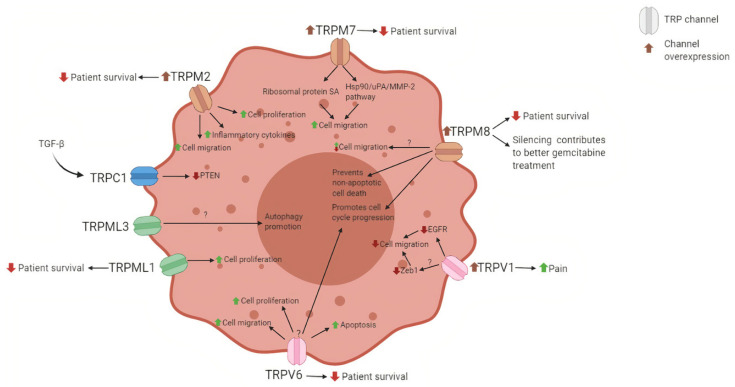
Summary of the roles and possible effects of the TRP channels in the pancreatic cancer cell. The TRP channels exert several changes to the PDAC cells. The expression of these channels correlates negatively with patient survival and may act on the level of cell proliferation, migration and apoptosis.

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
