# Peer review of "Role of the TRP Channels in Pancreatic Ductal Adenocarcinoma Development and Progression"

_cells, 2021, doi:10.3390/cells10051021_

Round 1

Reviewer 1 Report

Please check for the suggestions below:

Line 122, 123, 124 – please specify, which subfamily of TRPM do TRPM2, 7, 8, 4 and 6 belong?

Please differentiate between the Channel family, sub-family, and members of a sub-family by describing about them in sub-heading.

Line 128 – delete ‘moreover’

Line 198 – delete ‘Already’

Line 199, 200, 201 – Please rephrase the sentence “. Hartel et al. demonstrated the overexpression of the TRPV1 in patients 199 with both PDAC and chronic pancreatitis with a trend towards higher TRPV1 expression 200 in pancreatic cancer”

Line 220 – Genetic predisposition, delete ‘Genetical’

Line 282 – Delete ‘only the’ or rephrase the sentence as ‘Of these three, co-silencing of TRPC1 and NCX1 was able to inhibit TGF-β induced cell motility.

Line 292 – Instead of ‘open a bit the discussion on this channel’ please write ‘opened the discussion on this channel’.

Line 298 – Delete ‘As it happens with’ rephrase the sentence – ‘As observed with’ or ‘As in case of ‘ TRPC1 and TRPC3

Line 313 – Rephrase “Although many efforts have been made to understand and to treat PDAC” for example - ‘Despite of multiple efforts to understand and treat PDAC’

Line 324 – Rephrase “How can we apply the knowledge to the current treatments being applied?” for example – How can this knowledge contribute to the treatment or How can we translate this knowledge from bench-side to bed-side.

Line 325 – Rephrase “There is also 324 a gap in the knowledge of which channels compensate the lack of others?” for example – There is a gap in our understanding of redundancy among these channels.

Line 332 – Rephrase “Further studies should support this idea with more data..” for example as “Further study is needed to support the idea with more data…”

In the conclusion please incorporate your opinion regarding what can be done to improve the diagnosis and treatment for example in Line 335 – Please elaborate what kind of diagnostic tools are needed, Do you think – Mouse models, lineage tracing tools could be useful?

Figure 2 – As mentioned in the figure legend the figure is inspired by Clapham 2003 (see below) review article. In the Clapham 2003 the N and C termini of channels have been compared horizontally, which makes it easier for the reader to compare the difference between families. I would recommend Mesquita et al. to orient the N and C terminus domains horizontally.

I would also suggest to divide the Figure 2 into:

 Figure 2 (a) showing the 6 transmembrane domains, please mark the pore region between domain 5 and 6.

Figure 2 (b) showing the N and C termini domain of different families.

Please re-write the figure legend  for Figure 2 - (a) and (b) instead of writing it as a paragraph.

To describe the sub-families, please include a TRP family tree as done in the reference below: (https://www.guidetopharmacology.org/GRAC/FamilyIntroductionForward?familyId=78)

This figure was also modified from Clapham DE (2003) TRP channels as cellular sensors. Nature 426: 517-524 

Reviewer 2 Report

In their manuscript entitled "Role of TRP channels in pancreatic cancer development and progression," Mesquita et al. summarize the contribution of the TRP channel family members in pancreatic ductal adenocarcinoma (PDAC). Therefore, the title of the review should be changed since the review only highlights the role of TRP channels in one type of pancreatic cancer, PDAC. The abstract and introduction contain well-known information about TRP channels and PDAC. There are several grammatical errors in the manuscript which strongly disturb the flow of information. Although the authors listed several experimental results in the TRP chapters, the interpretation is often superficial and does not provide the analysis of TRP-mediated signaling in PDAC. In general, extensive language editing by a native English speaker is required before submitting a revised manuscript.

Specific comments and suggestions:

  1. The chapter of "Pancreatic Ductal Adenocarcinoma" is not focused, describing basic textbook knowledge that is irrelevant to understanding the link between aberrant Ca2+ homeostasis and TRP channel function and PDAC. Therefore the attention of the readership will be not be attracted. It should be focused on Ca2+ homeostasis in PDAC. In addition, the authors should include a couple of sentences on the composition of the tumor microenvironment in pancreatic cancer (which cellular components are important contributors to PDAC).

  1. Pancreatitis-PDAC transition can be triggered by shear-stress, mechanosensing or inflammation-mediated mechanisms. Although in chapter TRPV4, the authors indicate the risk of pancreatic cancer development from pancreatitis involving TRPV4, possible mechanisms were not discussed. Recently, Swain et al., 2020, JCI highlighted the role of TRPV4 in pancreatitis by Piezo channel activation. The authors should these pathophysiological aspects in more detail based on recent findings reported in the literature.

  1. Preceding the conclusion section, the authors should add one more chapter on the therapeutic potential of TRP targeting in PDAC. Here, the authors may want to discuss the biological rationale and future therapeutic perspectives for anti-TRP strategies in pancreatic cancer. The conclusion can be phrased in more general terms.

  1. The authors should rephrase or delete the following sentences:

Page 1. Line 10:  "From the moment of their discovery"

Page 1. Line 16: "As of now "

Page 1. Line 25: "fantastic discovery "

Page 1. Line 28-30: "The understanding of their homology sharing and the unreasonable amount of names given to the same proteins led to a standardization of the nomenclature in 2002. The term TRP superfamily was coined."

Page 1. Line 36: "These functions are further studied in the cited reviews".

Page 1. Line 36-37: "The role of these channels in diverse cancerous diseases has been established although the clinical output of such knowledge has not yet been seen".

Page 1. Line 39:" too many loose ends"

Page 2. Line 48. "Sadly, there has hardly been any development to the better during the last 5 years"

Page 2. Line 56. "utmost importance"

Page 2. Line 57. "For the first and last processes"

Page 2. Line 71-73. "in order to migrate away from the primary tumor and invade surrounding tissues cells need to establish contacts with the extracellular-matrix (ECM), with several signalling linkage points denominated focal adhesions"

Page 2. Line 7 6. "Concerning invasion"

Page 3. Line 94 "to surpass"

Page 4. Line 119. "biggest of the  family"

Page 5. Line 157. "TRPM8 was found overexpressed"

Page 5. Line 157. "TRPM8 was required  for maintaining proliferation by promoting survival"

Page 6. Line 190. "Although being present in the pancreatic tissue, TRPV2 is usually related to the pancreatic endocrine system"

Page 6. Line 200. "with a trend towards higher TRPV1 expression in pancreatic cancer"

Page 6. Line 215. "The KO of TRPV1 channels "

Page 7. Line 233. "In pancreatic cancer there is a bit of controversy on this regard".

Page 7. Line 239. "The silencing of TRPV6"

Page 7. Line 268. "However, so far"

Page 8. Line 292. " open a bit the discussion on this channel"

Page 8. Line 314. the overall survival of the patients is still very disappointing

Page 8. Line 315.  "there is a lot more knowledge to work with"

Page 9. Line 344. "there is a need to output these effects"

Round 2

Reviewer 2 Report

All points raised by the reviewer were addressed satisfactorily.